# New Insights into the Regulatory Role of Ferroptosis in Ankylosing Spondylitis via Consensus Clustering of Ferroptosis-Related Genes and Weighted Gene Co-Expression Network Analysis

**DOI:** 10.3390/genes13081373

**Published:** 2022-07-31

**Authors:** Tianhua Rong, Ningyi Jia, Bingxuan Wu, Dacheng Sang, Baoge Liu

**Affiliations:** 1Department of Orthopaedic Surgery, Spine Center, Beijing Tiantan Hospital, Capital Medical University, Beijing 100070, China; rongtianhua@163.com (T.R.); bingxuanwoo@163.com (B.W.); dachengsang1616@163.com (D.S.); 2National Clinical Research Center for Orthopedics, Sports Medicine & Rehabilitation, Beijing 100070, China; 3Basic and Translational Medicine Center, China National Clinical Research Center for Neurological Diseases, Beijing 100070, China; 4Department of Obstetrics and Gynecology, Beijing Tiantan Hospital, Capital Medical University, Beijing 100070, China; bonniejianingyi@sina.com

**Keywords:** ankylosing spondylitis, ferroptosis, consensus clustering, weighted gene co-expression network analysis, hub genes, regulatory networks

## Abstract

**Background**: The pathogenesis of ankylosing spondylitis (AS) remains undetermined. Ferroptosis is a newly discovered form of regulated cell death involved in multiple autoimmune diseases. Currently, there are no reports on the connection between ferroptosis and AS. **Methods**: AS samples from the Gene Expression Omnibus were divided into two subgroups using consensus clustering of ferroptosis-related genes (FRGs). Weighted gene co-expression network analysis (WGCNA) of the intergroup differentially expressed genes (DEGs) and protein–protein interaction (PPI) analysis of the key module were used to screen out hub genes. A multifactor regulatory network was then constructed based on hub genes. **Results**: The 52 AS patients in dataset GSE73754 were divided into cluster 1 (*n* = 24) and cluster 2 (*n* = 28). DEGs were mainly enriched in pathways related to mitochondria, ubiquitin, and neurodegeneration. Candidate hub genes, screened by PPI and WGCNA, were intersected. Subsequently, 12 overlapping genes were identified as definitive hub genes. A multifactor interaction network with 45 nodes and 150 edges was generated, comprising the 12 hub genes and 32 non-coding RNAs. **Conclusions**: AS can be divided into two subtypes according to FRG expression. Ferroptosis might play a regulatory role in AS. Tailoring treatment according to the ferroptosis status of AS patients can be a promising direction.

## 1. Introduction

Ankylosing spondylitis (AS) is a radiographic form of axial spondyloarthritis, which also belongs to the broader category of seronegative spondyloarthropathy [1,2]. The approximate prevalence of AS varies from 0.1 to 0.3% between continents and occurs predominantly in men (with male/female ratios ranging from 2.3:1 to 3.8:1) [3]. The clinical manifestations of AS are back pain, enthesitis with syndesmophytes, loss of spinal mobility, and spinal deformity [4,5].

AS is a chronic inflammatory disorder, and its etiopathogenesis and natural course are closely related to the functional status of the immune system [6,7]. Immune activation, inflammatory response, and new bone formation may be involved in spondyloarthritis [8,9,10]. However, the molecular regulatory network linking inflammation and bone metabolism remains unclear. Therefore, uncovering the pathogenesis of AS is vital in the prevention of radiographic progression and secondary structural damage to the axial skeleton [5,11].

Ferroptosis, a newly discovered form of regulated cell death (RCD) characterized by iron-dependent lipid peroxidation [12], is regarded as a pervasive, disease-relevant metabolic RCD pathway [13]. Recent studies have increasingly reported on complex associations between ferroptosis and the immune system [14]. The regulatory activity of ferroptosis in immune function and inflammation is multifaceted and involves innate, acquired, and autoimmunity [15,16,17]. Therefore, it can be logically inferred that an etiological link may exist between ferroptosis and AS.

Although there are no reports on ferroptosis in AS pathogenesis, some indirect evidence supports this connection. Abnormal iron homeostasis occurs in rheumatic disease [18]. More specifically, iron overload, anemia, and altered ferritin levels are observed in AS [18,19,20]. Decreased levels of plasma thiol and antioxidant vitamins [21,22], increased lipid peroxidation, and protein oxidation [22,23,24,25] in AS patients have been described. These findings correspond to the three pillars of ferroptosis, which are iron, thiols, and lipid peroxidation, collectively suggesting that ferroptosis participates in AS pathogenesis [26].

To investigate the etiological role of ferroptosis in AS and explore the underlying molecular regulatory network, we analyzed AS microarray data obtained from the Gene Expression Omnibus (GEO) database. AS samples were clustered into two subgroups according to the expression patterns of ferroptosis-related genes. We then analyzed differential gene expression profiles between the two groups to identify critical pathways, co-expression networks, and hub genes. Based on the hub genes, we constructed a multifactor regulatory network comprising micro RNAs (miRNAs), long non-coding RNAs (lncRNAs), and transcription factors (TFs), thereby providing new insights into the molecular pathogenesis of AS.

## 2. Materials and Methods

### 2.1. Data Collection

First, we searched the GEO database (https://www.ncbi.nlm.nih.gov/geo/, accessed on 11 June 2021) [27] using the following terms: “ankylosing spondylitis”, “axial spondyloarthritis”, and “seronegative spondyloarthropathies”. Among the 14 retrieved results, 6 datasets that included analysis of peripheral blood samples obtained from AS patients were evaluated in detail. After excluding datasets with insufficient sample sizes and those lacking demographic information, the microarray dataset GSE73754 [28] was used for further analysis. All of the AS cases in this dataset had radiographic sacroiliitis as defined by the modified New York criteria for AS, and were evaluated using the Bath Ankylosing Spondylitis Disease Activity Index [28]. Gene expression profiles, shown as log2-transformed quantile-normalized signal intensities, were downloaded and preprocessed as follows: (1) all of the probes were mapped to genes; (2) probes that showed no signal were removed; and (3) when multiple probes corresponded to the same gene, the final expression level of a gene was determined using the arithmetic median of multiple probes. The present study was conducted in accordance with the ethical standards of the Institutional Review Board of our hospital.

### 2.2. Ferroptosis-Related Genes

Ferroptosis-related genes (FRGs) were compiled from an online database and published articles. FerrDb (http://www.zhounan.org/ferrdb, accessed on 11 June 2021) [29], a database for regulators and markers of ferroptosis, summarizes annotations on 259 FRGs from previous reports. Through a literature review, an additional 41 FRGs were curated after removing duplication [30,31,32]. Finally, a total of 300 FRGs (Appendix A) were utilized for subsequent bioinformatic analysis (Figure 1).

### 2.3. Consensus Clustering

Consensus clustering (CC), a data mining technique for the detection of unknown subgroups in a dataset, is widely used in functional genomic studies [33]. Unsupervised cluster discovery was used to analyze data on the 52 AS patients, and the results were visualized using the “ConsensusClusterPlus” package [34] in R 4.1.0 according to the expression levels of the above-described FRGs. The clustering algorithm was also applied to an RNA sequencing dataset GSE141646 [35] that contained 22 AS patients to verify the robustness of the clustering results.

### 2.4. Estimation of Immune Cell Type Fractions

The normalized gene-expression profiles with standard annotation were uploaded to the CIBERSORT online platform (http://cibersort.stanford.edu/, accessed on 30 July 2021). CIBERSORT is a deconvolution algorithm for characterizing the cellular composition of complex tissues [36]. The algorithm was run using the “LM22” signature matrix of 547 genes at 1000 permutations to estimate the relative fractions of 22 immune cell types. All of the samples had a CIBERSORT output of *p* < 0.05, indicating sufficient accuracy of the inferred fractions of immune cell populations. For each AS sample, the sum of all of the evaluated immune-cell fractions was normalized to one. Fractions were illustrated using patient clusters in stacked bar plots and box plots constructed using the “ggpubr” package.

### 2.5. Screening for Differentially Expressed Genes

Inter-cluster statistical comparisons of gene-expression levels were performed using in-built functions of R. Student’s *t*-test and Wilcoxon–Mann–Whitney test were used to analyze normally and non-normally distributed data, respectively [37]. To control for false discovery rate (FDR), *p*-values were adjusted for multiple comparisons using the Benjamini–Hochberg method [38]. An adjusted *p* value < 0.05 was defined as the threshold for screening differentially expressed genes (DEGs). DEGs were visualized using volcano plots, heatmap, and principal component analysis (PCA) in the R packages “ggpubr,” “pheatmap”, and “factoextra”, respectively.

### 2.6. Gene Ontology and Pathway Enrichment Analysis

Enrichment analysis of Gene Ontology (GO) categories and Kyoto Encyclopedia of Genes and Genomes (KEGG) pathways associated with the DEGs examined in our present study was performed using the “clusterProfiler” package in R [39]. The top 10 most significantly enriched GO terms and KEGG pathways were visualized using the “ggplot2” R package.

### 2.7. Weighted Gene Co-Expression Network Analysis

Weighted gene co-expression network analysis (WGCNA) is an algorithm that assesses the relationships between measured transcripts, identifies clinically relevant co-expressed gene modules, and explores key genes in disease pathways from the perspective of systems biology [40]. The WGCNA R software package was used to construct AS-associated modules from the DEGs described above. To achieve a scale-free network, optimal soft thresholding power β for increasing expression similarity and calculating adjacency was determined using a “pickSoftThreshold” function within the package. Next, the gene correlation matrix was transformed into an adjacency matrix, which was further converted into an unsigned topological overlap matrix (TOM). According to TOM, average-linkage hierarchical clustering was used to obtain DEG clusters and construct dendrograms. Using a minimum module size of 30 genes, the dynamic tree cut algorithm (deepSplit = 2) was used to determine gene modules; DEGs having similar expression patterns were designated into the same modules. Module eigengenes (MEs) were calculated as the first principal component of expression profiles in each module. Modules were then clustered and merged according to ME dissimilarities (mergeCutHeight = 0.25). Correlations between MEs and clinical traits of AS patients were calculated using Pearson’s correlation coefficient. Then, the module having the highest coefficient (key module) was targeted, and intra-modular genes were extracted for further analyses.

### 2.8. Identification of Hub Genes in AS

Intra-modular connectivity of each gene in the key WGCNA module was expressed as a measure of module membership (MM), and the AS-related biological correlation was evaluated using gene significance (GS) [40]. MM ≥ 0.7 and GS ≥ 0.5 were defined as thresholds for screening intra-modular hub genes. In addition, all of the genes in the key module were uploaded into the STRING online database (version 11.5, https://string-db.org/, accessed on 30 July 2021) for the prediction of protein–protein interactions (PPI) [41]. Interactions having a confidence score > 0.9 were imported into Cytoscape (version 3.8.2) to construct and visualize the PPI network [42]. In this PPI network, nodes having increased connectivity were regarded as hub genes, which were identified using MCODE and CytoHubba plug-ins in Cytoscape. Hub genes identified using WGCNA and PPI were intersected to obtain the AS-related final hub genes, which were then represented using a Venn diagram in the “VennDiagram” package in R.

### 2.9. Construction of Regulatory Network Based on Hub Genes

Differences in hub gene expression levels between clusters were validated using the “ggpubr” in the R package. Pairwise expression correlations between hub genes were analyzed using “corrplot” in the R package. Then, manually curated human transcriptional regulatory networks between TFs and target genes (TF–mRNA) were downloaded from the TRRUST database (version 2, https://www.grnpedia.org/trrust/, accessed on 30 July 2021). The current version of TRRUST contains 8444 and 6552 TF–target regulatory relationships of 800 human TFs and 828 mouse TFs, respectively. They have been derived from 11,237 Pubmed articles, which describe small-scale experimental studies of transcriptional regulations [43]. Interactome data, including mRNA–lncRNA and mRNA–miRNA interaction pairs, were accessed using the RNAInter database (version 4.0, http://www.rnainter.org/, accessed on 30 July 2021). RNAInter integrates experimentally validated and computationally predicted RNA interactome data from the literature and databases. It is featured with a redefined confidence scoring system and an update of entries to over 29 million interactions in Homo sapiens [44]. After data integration, a multifactor regulatory network involving non-coding RNAs, TFs, and mRNAs of AS-involved hub genes was constructed and visualized using Cytoscape to explore the molecular regulatory mechanism of AS in the context of ferroptosis.

## 3. Results

The dataset GSE73754 obtained from GEO contained 72 samples collected from 52 AS patients and 20 healthy controls. Microarray expression profiling was performed using whole-blood RNA analysis performed on an Illumina HumanHT-12 V4.0 expression beadchip. We annotated the probes with gene symbols and tried to locate the 300 manually compiled FRGs in GSE73754. After a thorough search, 276 out of the 300 FRGs were identified in this dataset. The other 24 FRGs were not contained by the present microarray. The age, sex, and HLA-B27 status of each participant were provided for each sample. The expression profiles of the 276 FRGs were plotted using heatmaps. No robust pattern of FRG differential expression was identified when analyzed by age, sex, or AS status (Figure 1 and Appendix A).

### 3.1. Consensus Clustering Based on FRGs

Based on expression-level similarity, high intra- and low inter-class correlation, and clinical interpretability of the 276 FRGs, the optimal number of clusters (k value) in our consensus matrix was defined as 2, with the area under empirical cumulative distribution function increasing and delta area decreasing from k = 2 to 6 (Figure 2A,B) [33]. The consensus clustering algorithm categorized samples with similar expression patterns of the 276 FRGs into the same cluster. Therefore, the 52 AS patients were divided into cluster 1 (*n* = 24) and cluster 2 (*n* = 28), after which the clinical characteristics of the patients were compared between the two clusters (Figure 2C,D and Appendix A). There was no inter-cluster statistical difference in age, sex, or HLA-B27 status of the patients. The same clustering trend (k = 2) was identified in the external validation dataset GSE141646 [35] (Appendix A).

### 3.2. Composition of Immune Cells

Leukocyte deconvolution was successfully performed for all of the AS samples using the CIBERSORT computational approach (Appendix A). Among the 22 hematopoietic cell phenotypes, neutrophils, CD8^+^ T cells, monocytes, and resting natural killer (NK) cells were the four most commonly occurring cell types, which was consistent with the composition of human peripheral blood. Immune cell fractions are illustrated in a stacked bar plot shown in Figure 3A and a box plot shown in Figure 3B. Wilcoxon signed-rank test indicated that, in terms of innate immunity, cluster 2 contained significantly higher proportions of neutrophils, activated dendritic cells, and resting mast cells compared with those of cluster 1; in terms of the adaptive immune system, cluster 1 contained a significantly higher proportion of CD8^+^ T cells compared with that of cluster 2 (*p* < 0.05).

### 3.3. Identification of DEGs between the Two Clusters

After the two clusters were obtained by consensus clustering algorithm based on the 276 FRGs, our scope of analysis shifted to the entire microarray. The expression levels of all genes in the microarray were compared between cluster 1 and cluster 2, and a total of 3663 DEGs were screened out. These DEGs included 2428 upregulated and 1235 downregulated genes (Figure 4A,B). When plotted using the first two principal components, no statistical difference was found between the two clusters in the rectangular coordinate system (Figure 4C), which necessitated further systematic analysis, namely the WGCNA below. Among the 3663 DEGs, 86 genes belonged to the above-mentioned 276 FRGs and included 46 drivers, 25 suppressors, and 31 markers (Figure 4D and Appendix A), confirming the efficiency of the CC algorithm.

### 3.4. Functional Annotation of DEGs

Our GO analysis indicated that the top three enriched biological processes for DEGs were ribonucleoprotein complex biogenesis, ncRNA metabolic process, and ncRNA processing. The significantly enriched cellular component terms included mitochondrial inner membrane, mitochondrial matrix, and ribosome. The three most enriched molecular functions were structural components of the ribosome, catalytic activity acting on RNA, and ubiquitin-like protein ligase binding (Figure 5A). The results of the KEGG analysis indicated that DEGs were mainly enriched in pathways involving neurodegeneration-multiple diseases, amyotrophic lateral sclerosis, and ribosome (Figure 5B).

### 3.5. WGCNA and the Key Module

WGCNA was used to analyze DEGs between the two FRG-based clusters in the 52 AS samples. The power of soft thresholding was 16 and the scale-free fit index (signed R^2^) was 0.87 (Figure 6A,B). A total of five co-expression modules were constructed according to our DEG hierarchical clustering dendrogram (Figure 6C). The heatmap analysis of the correlations between module eigengenes and clinical traits indicated that the turquoise module was most significantly correlated with AS, and was thereby defined as the key module containing 1911 DEGs (52.17% of all DEGs, 1911/3663) (Figure 6D and Appendix A).

### 3.6. Identification of Hub Genes

According to our screening criteria of MM ≥ 0.7 and GS ≥ 0.5, a total of 34 intramodular hub genes were extracted from the WGCNA turquoise module (Figure 7A). In the PPI network constructed using STRING, 317 of the 1911 DEGs were identified as candidate hub genes. A hub gene-based PPI subnetwork was then excavated, and only the edges representing the highest confidence (interaction score > 0.9) were drawn (Figure 7B). The candidate hub genes screened using PPI and WGCNA were intersected. Subsequently, 12 overlapping genes were identified as definitive hub genes (Figure 7C). None of these 12 hub genes belonged to the above-mentioned 276 FRGs. Pairwise expression correlation analysis indicated consistently positive correlations between any 2 of the 12 hub genes (Figure 7D). In addition, the expression levels of the 12 hub genes were significantly higher in cluster 1 than those in cluster 2 (Figure 8).

### 3.7. Multifactor Regulatory Network in AS

First, 8444 human TF–target regulatory relationships were obtained from the TRRUST v2 database, and then complete RNA–RNA interactions with confidence scoring were downloaded from the RNAInter v4.0 database. TFs, miRNAs, and lncRNAs showing interaction with the 12 hub genes were screened, and then lncRNAs communicating with these miRNAs were selected. The number of regulatory pairs was trimmed for layout optimization, and an interaction score >0.2 was set as the threshold for edges between RNA nodes [44]. Finally, a multifactor regulatory network containing 45 nodes and 150 edges was generated (Figure 9A). The top six most connected regulatory factors (degree > 6) included four lncRNAs (SNHG16, TUG1, MIR17HG, and GAS5) and two miRNAs (hsa-miR-106a-5p and hsa-miR-18a-5p) (Appendix A). Furthermore, a competing endogenous RNA (ceRNA) subnetwork that included only nodes with connectivity >5 was extracted; this subnetwork may serve as a molecular regulatory bridge between AS and ferroptosis (Figure 9B).

## 4. Discussion

AS, a refractory autoimmune disease that affects axial bones and joints, is highly teratogenic and disabling in patients, posing a formidable challenge to clinicians and researchers [45,46]. Uncovering the mechanisms driving AS pathogenesis is critical in the development of novel therapeutics for this patient population. Although numerous etiological studies on AS have been performed, the osteoimmunological pathogenesis of AS and relevant therapeutic targets remain undetermined [10,47]. Previous studies have shown that ferroptosis is closely related to bone metabolism and the immune system [14,16,17,48,49]. However, the role of ferroptosis in the development and progression of AS has not been previously documented. To the best of our knowledge, this study is the first to show a potential connection between ferroptosis and AS via bioinformatics analysis. Our primary findings involved two aspects: characterization of two FRG-related subtypes of AS patients, and construction of an AS regulatory network based on the 12 hub genes identified in our present study.

The role of ferroptosis in musculoskeletal diseases has been well documented [50]. Ferroptosis in chondrocytes leads to progression of osteoarthritis, which can be alleviated by inhibiting chondrocyte ferroptosis [51,52]. Studies using nucleus pulposus cells from both human and animal models have shown that ferroptosis is involved in intervertebral disc degeneration [53,54,55]. Ding et al. [56] and Huang et al. [57] suggested that targeting ferroptosis might be used in the treatment of patients with sarcopenia. Ferroptosis also regulates the balance between osteoblasts and osteoclasts, and participates in the development of osteoporosis [48,49,58]. Musculoskeletal manifestations, including inflammatory bone destruction and heterotopic ossification, are hallmarks of AS. Therefore, these previous findings on ferroptosis-associated musculoskeletal disorders have led us to hypothesize that ferroptosis may also play a vital role in the development and progression of AS. Previous bioinformatics studies have provided data supporting our hypothesis. Zhang et al. compared gene expression profiles between male AS patients (*n* = 27) and matched healthy controls (*n* = 10) in GSE73754, and identified that key genes in AS pathogenesis also regulated immune-cell functions [59]. That study also showed that the ferroptosis pathway is a significantly enriched term through a gene set enrichment analysis of KEGG pathways. Meng et al. performed a comparative bioinformatics analysis in 16 AS patients and 16 matched controls using the dataset GSE25101, and found that ferroptosis is one of the significantly enriched terms in the KEGG analysis of upregulated DEGs [60]. Based on this evidence, we used computational methods to further investigate the molecular mechanisms connecting ferroptosis and AS in order to provide a basis for future etiological studies on AS, and aid in the development of new therapeutic targets for AS patients.

Positivity for human leukocyte antigen class I molecule B27 (HLA-B27) has long been associated with AS [61]. The underlying pathogenic mechanisms include the presentation of arthritogenic peptides, the unfolded protein response and the endoplasmic reticulum stress response, and the proinflammatory effects of cell surface HLA-B27 free heavy chain [61]. Because variants within the HLA-B27 region can lead to the presentation of arthritogenic peptides to CD8^+^ cytotoxic T cells, CD8^+^ T cells are traditionally thought to be implicated in AS pathogenesis [7,62,63]. Several studies have suggested that activation of CD8^+^ T cells is regulated by ferroptosis, and may enhance ferroptosis-specific lipid peroxidation in surrounding tumor cells during immunotherapy [16,64,65]. In our present study, we used CIBERSORT analysis to identify a significant difference in the proportions of CD8^+^ T cells between the two FRG-related clusters, which agreed with the findings obtained in previous studies. Neutrophils, another type of immune cells, also showed an inter-cluster difference in fractions, echoing recent findings showing that innate immunity is also important in mediating AS and ferroptosis [14,66]. Patients with AS show a high degree of infiltration by IL-17^+^ neutrophils in facet joints and an elevated neutrophil–lymphocyte ratio in peripheral blood [67,68]. The presence of neutrophil extracellular traps (NETs) in AS is associated with differentiation of mesenchymal stem cells (MSCs) toward bone-forming cells [69]. In terms of ferroptosis, current evidence indicates that lipid peroxidation serves as an upstream signal of NET-induced ferroptosis [70], that neutrophils participate in ferroptotic tissue damage [71], and that neutrophil ferroptosis can induce systemic autoimmunity [17]. In summary, our findings and those obtained in previous studies synergistically support the pivotal role of ferroptosis in the pathogenesis of AS in the context of the immune system.

The keywords obtained using enrichment analysis mainly centered on the following concepts: structure and function of mitochondria and ribosomes, ubiquitin, RNA metabolism, and neurodegeneration. Ribosomes and RNA are fundamental to nearly all biological processes and are not linked to certain clinical traits; thus, they are beyond the scope of this discussion. Mitochondria are a major source of intracellular reactive oxygen species (ROS) and lipid peroxides, which play a crucial role in regulating multiple types of RCD [72]. Although it seems somewhat paradoxical, recent studies have suggested that mitochondria have both pro-and anti-ferroptosis functions owing to the diversity of their ferroptosis-related metabolic activities [73,74,75]. Imbalanced redox and altered glutamine metabolism, which are observed in patients with AS, are closely related to mitochondrial dysfunction [22,23,24,25,76]. In addition, Ye et al. reported that serum with increased levels of oxidative stress markers obtained from AS patients promotes senescence in MSCs by mediating mitochondrial dysfunction and excessive ROS production [77]. Therefore, mitochondrial pathways may be the common mechanism shared by AS and ferroptosis; however, this notion warrants further studies to better elucidate the etiological role of ferroptosis in AS. Although the role of ferroptosis in neurodegeneration has been extensively investigated, data on the connection between AS and neurodegenerative diseases remain limited. Epidemiological studies have reported hazard ratios ranging from 1.75 to 1.82 (*p* < 0.001) for the development of Parkinson’s disease (PD) and parkinsonism in AS population [78,79]. Mitochondrial dysfunctions, particularly abnormal oxidative phosphorylation, and epithelial barrier disruption in the gut may be the mechanisms underlying the clinical association between AS and PD [59,80,81]. In addition, complex interactions between the ubiquitin-proteasome system and cellular degradation, which participate in autophagy-dependent ferroptosis and progression of AS, may be another possible link between ferroptosis and AS [82,83,84,85].

In general, data obtained using microarray expression profiling are more complex, unlike a simple list of screened-out DEGs. Hence, to fully exploit the interrelationships between the expression levels of all profiled genes, we further analyzed the GSE73754 dataset using CC and WGCNA [34,40]. Instead of comparing AS patients with healthy controls, we isolated two clusters within the AS group; these two clusters showed distinct FRG expression patterns. Comparative analyses of the two FRG-related subtypes of AS provided preliminary support for our hypothesis stating that ferroptosis participates in AS pathogenesis. Based on 3663 inter-cluster DEGs, we constructed five co-expression modules, with each module containing a group of biologically related genes. After calculating the eigengene significance, the turquoise module showed the highest correlation with AS and was defined as the key module. Among the 1911 genes in this key module, 12 hub genes were finally obtained by intersecting the results of WGCNA analysis with those of PPI. A regulatory network containing these hub genes, miRNAs, and lncRNAs was then constructed. Within these hub genes, *SUMO2* had the greatest number of related references. *SUMO2* encodes small ubiquitin-like modifier 2 protein, which participates in post-translational protein modification; functions in a ubiquitin-like manner; and regulates multiple biological processes including bone metabolism, oxidative stress, autophagy, and immune homeostasis [86,87,88,89]. These findings suggest that *SUMO2* plays an important role in the regulation of AS-related ferroptosis. The hub gene *NDUFS4* was described in two previous bioinformatics AS studies that analyzed a GEO dataset GSE25101, serving as external validation [80,90]. *NDUFS4* encodes an accessory subunit of mitochondrial complex Ⅰ, a major component in the mitochondrial respiratory chain; this accessory subunit of mitochondrial complex I is also crucial in the reconciliation between innate immunity and skeletal homeostasis [91]. The role of *NDUFS4* in ferroptosis, however, requires further investigation. The functions of the other 10 hub genes were consistently related to transcriptional regulation or mitochondrial function [92], but whether they are pivotal regulators of the associations between ferroptosis and AS remains to be determined.

Numerous non-coding RNAs are involved in AS pathogenesis [93]. In our present study, the ceRNA subnetwork analysis indicated that two miRNAs showed higher degrees of centrality compared with those of other miRNAs. miR-18a-5p and miR-106a-5p, which have not been described in existing studies on AS, are jointly associated with acute heart failure [94]. miR-18a-5p plays a crucial role in apoptosis and ROS production mediated by ischemia-reperfusion injury [95]. Together with lncRNA GAS5 and FENDRR, miR-18a-5p also participates in the regulation of mitochondrial ROS generation and energy metabolism [96,97]. Zhang et al. reported that the ceRNA axis circRHOT1/mir-106a-5p/STAT3 regulates ferroptosis in breast cancer [98]. Moreover, miR-106a-5p is involved in the progression of rheumatoid arthritis [99] and pathogenesis of adolescent idiopathic scoliosis [100], and regulates the differentiation of human osteoclasts via ceRNA networks [101]. Among lncRNAs in the subnetwork examined in our present study, TUG1 shows downregulated expression in AS [102]. SNHG16 is related to *MAPK1*, which is one of the key genes in ferroptosis; SNHG16 is also involved in intracerebral hemorrhage [103]. GAS5 has been described in both an AS cross-sectional study and a ferroptosis-related bioinformatics study [104,105]. Because these non-coding RNAs are simultaneously associated with redox homeostasis and musculoskeletal disorders, they may be bridging molecules between ferroptosis and AS, and may, therefore, be possible therapeutic targets and tools in future research. Further in-depth studies are needed to clarify the complex regulatory roles of ferroptosis in the development and progression of AS.

The primary limitation of our present study was the lack of experimental validation. Although the bioinformatics algorithms used in our analyses provided essential biological significance to the computational results, the molecular mechanisms of the 12 hub genes need to be further examined in well-designed future studies. Second, our present study employed limited data sources, and only one eligible dataset was enrolled, resulting in a relatively limited sample size. Thirdly, gene expression levels were influenced by AS status and a variety of confounding factors, hence the basal clinical and demographic data derived from the analyzed dataset were relatively insufficient. Comprehensive integration of clinical characteristics and omics data is recommended in future studies. Additionally, the lack of investigation into the IL-17 pathway was also a limitation of this study.

## 5. Conclusions

In conclusion, patients with AS can be divided into two subtypes using FRG-based consensus clustering analysis. The two subgroups showed distinct FRG expression patterns, different proportions of immune cell types, and DEGs enriched in mitochondria- and ubiquitin-related pathways. Using WGCNA analysis of intergroup DEGs, 12 hub genes were identified and a multifactor regulatory network was constructed. According to the existing literature, the highly connected nodes in the network were closely related to both redox homeostasis and the musculoskeletal system. The results obtained in our present study indicate that ferroptosis may play a significant role in the pathogenesis and molecular regulation of AS. Tailoring therapeutic protocols according to the ferroptosis status of each AS patient may present a feasible strategy in the treatment of AS.

## Figures and Tables

**Figure 1 genes-13-01373-f001:**
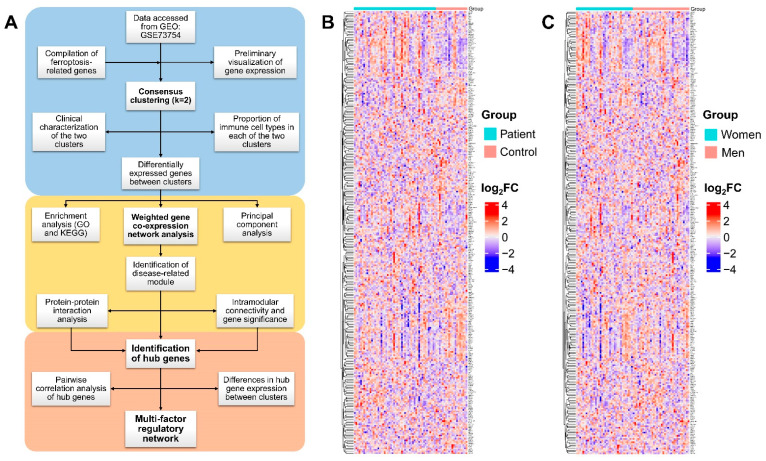
(**A**) Workflow of data processing and bioinformatics analysis, comprising three main modules, i.e. consensus clustering, weighted gene co-expression network analysis, and obtaining of hub genes and regulatory network. (**B**,**C**) Heatmaps of 276 ferroptosis-related genes plotted by AS status and sex, respectively. Red indicates high expression and blue indicates low expression. GEO, Gene Expression Omnibus; GO, Gene Ontology; KEGG, Kyoto Encyclopedia of Genes and Genomes; FC, fold change.

**Figure 2 genes-13-01373-f002:**
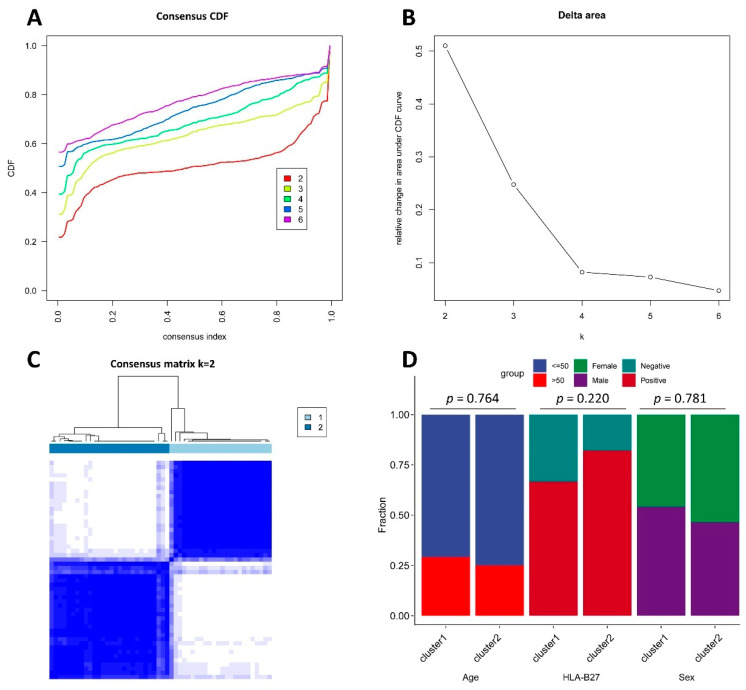
Visualization of consensus clustering. (**A**,**B**) Empirical consensus CDF plots and delta area score plots for k = 2 to 6, indicating that k = 2 or 3 is acceptable. (**C**) Consensus score matrix for the 52 AS samples obtained from GSE73754 when k = 2. Given our sample size and interpretability, the optimal k value was set as 2. (**D**) Percentage bar chart showing age, HLA-B27 status, and sex distribution in the two clusters. No statistical difference was identified. CDF, cumulative distribution function.

**Figure 3 genes-13-01373-f003:**
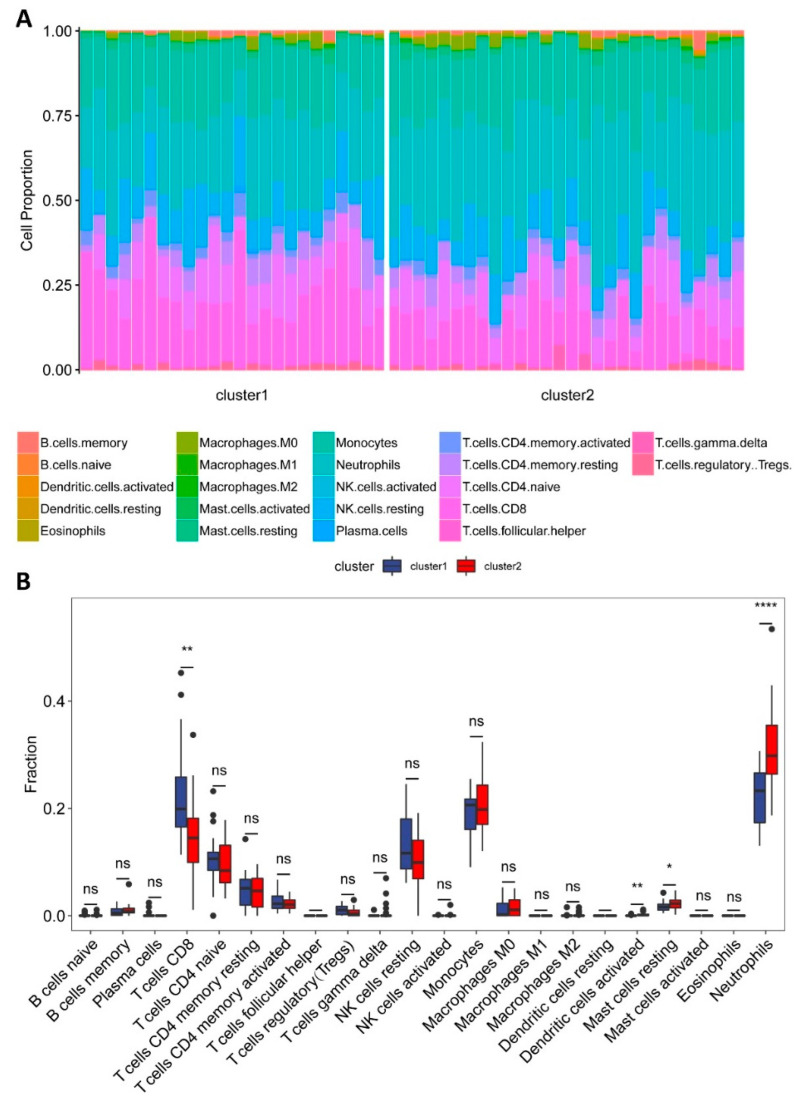
Immune cell type deconvolution. (**A**) Stacked bar plots showing proportions of immune cells in peripheral blood, separated by the two clusters. Each type of immune cell is represented by a different color. (**B**) Box plots illustrating differences in proportions of 22 immune cells between the two clusters. The asterisks indicate that the differences are statistically significant. * *p* < 0.05, ** *p* < 0.01, **** *p* < 0.0001. ns, not significant.

**Figure 4 genes-13-01373-f004:**
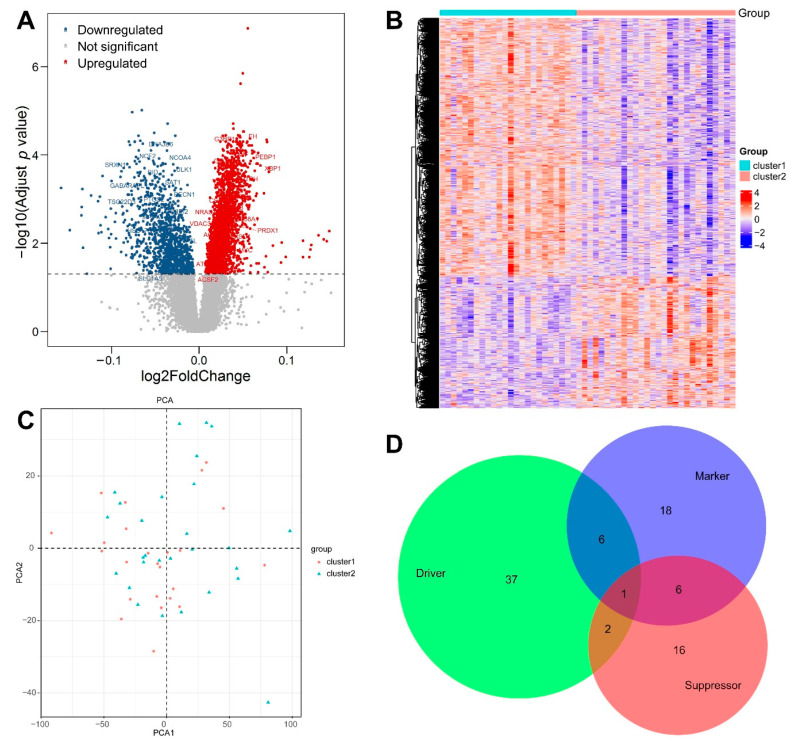
(**A**) Volcano plot of the differentially expressed genes between the two clusters, with some frequently reported ferroptosis-related genes being labeled. The cut-off adjusted *p*-value was <0.05. (**B**) Heatmap of the differentially expressed genes between the two clusters. Red indicates high expression and blue indicates low expression. (**C**) Scatter plot of the principal component analysis (PCA). The *x*-axis and *y*-axis refer to the first and second principal components, respectively. (**D**) Venn diagram of the 86 differentially expressed ferroptosis-related genes between the two clusters, with annotations from the FerrDb database.

**Figure 5 genes-13-01373-f005:**
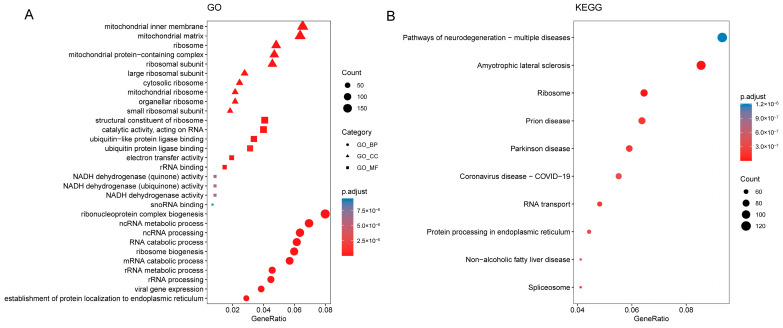
Enrichment analysis of the differentially expressed genes between the two clusters, conducted using Gene Ontology (GO) and Kyoto Encyclopedia of Genes and Genomes (KEGG). (**A**) The top 10 enriched GO terms in cellular components, molecular functions, and biological processes, respectively. (**B**) The top 10 enriched KEGG pathways. The warmer color represents a higher statistical significance.

**Figure 6 genes-13-01373-f006:**
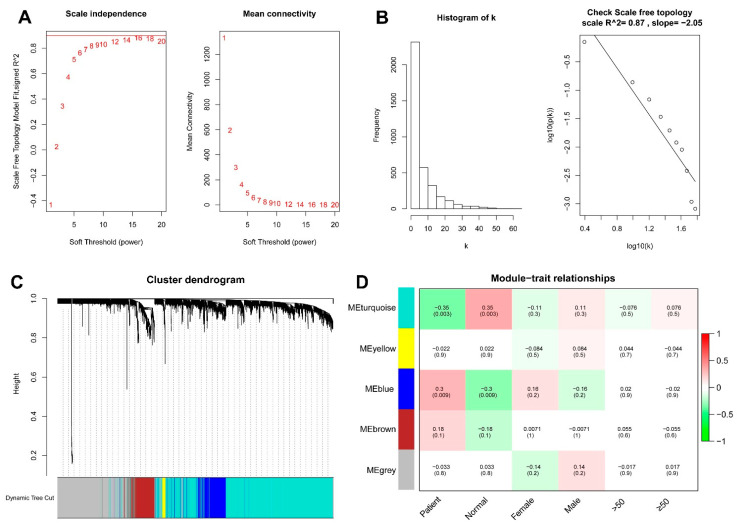
Weighted gene co-expression network analysis of differentially expressed genes between the two clusters. (**A**) Analysis of scale-free fit index and mean connectivity for various soft-thresholding powers (β value on *x*-axis). (**B**) Validation of scale-free property of the topology matrix. The variable k refers to the connectivity of each node. (**C**) Cluster dendrogram of co-expressed genes shows the construction of co-expression modules. (**D**) Heatmap of the module–trait relationships with corresponding correlation coefficients and *p*-values.

**Figure 7 genes-13-01373-f007:**
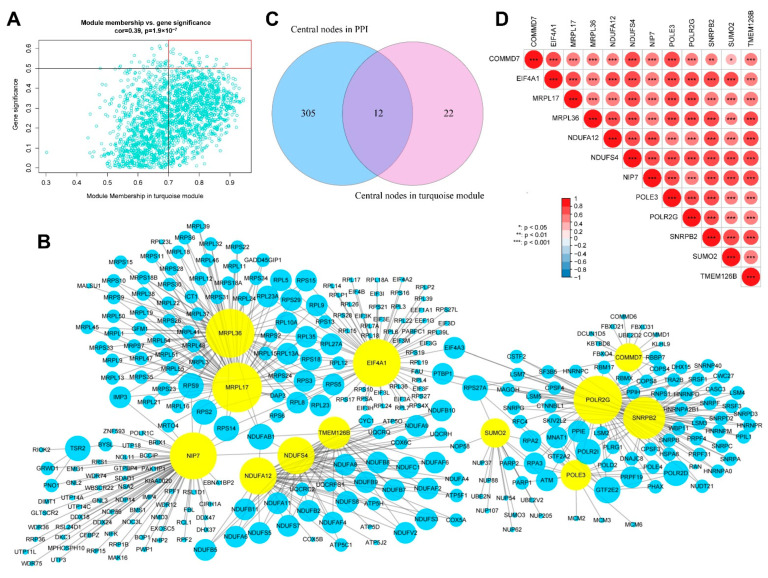
Identification of hub genes. (**A**) Scatter plot of module membership and gene significance in the turquoise module in weighted gene co-expression network analysis (WGCNA). The 34 genes in the upper right area are candidate hub genes. (**B**) Protein–protein interaction (PPI) network of the 317 candidate hub genes identified using Cytoscape. Yellow nodes represent the 12 selected AS-related hub genes. (**C**) Venn diagram shows the intersection between candidate hub genes in WGCNA (blue circle) and PPI (purple circle). (**D**) Pairwise correlation analysis of the 12 hub genes shows significant positive correlations between each gene.

**Figure 8 genes-13-01373-f008:**
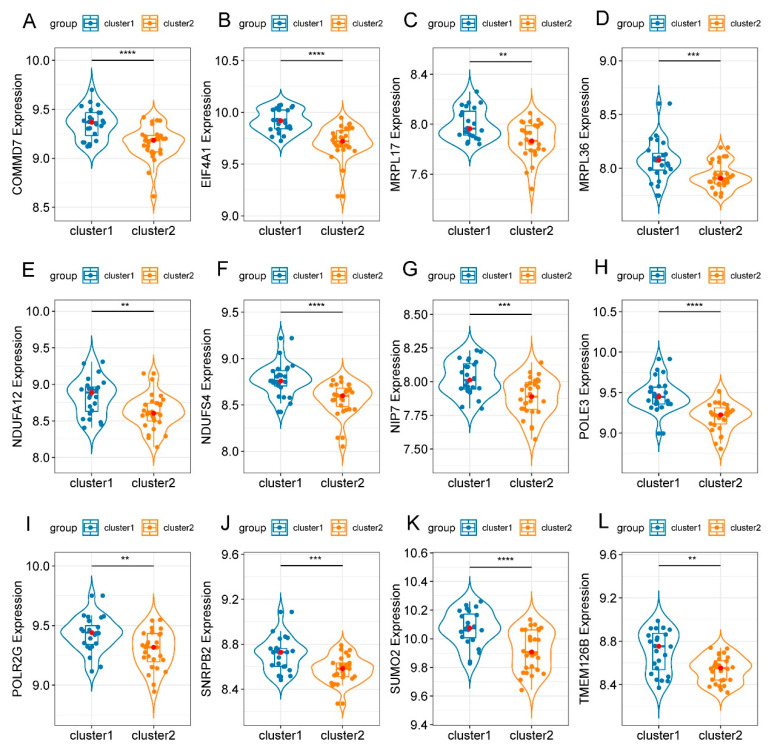
Violin diagrams demonstrate the expression levels of hub genes in the two clusters. (**A**–**L**) The expression levels of the 12 hub genes were consistently higher in cluster 1 than in cluster 2. The asterisks indicate that the differences are statistically significant. ** *p* < 0.01, *** *p* < 0.001, **** *p* < 0.0001.

**Figure 9 genes-13-01373-f009:**
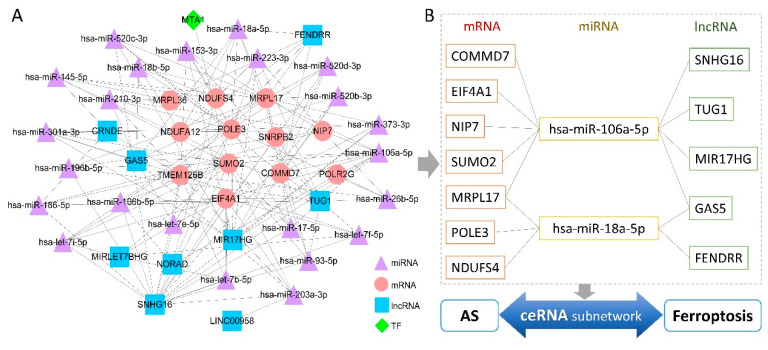
(**A**) Multifactor regulatory network based on the 12 hub genes. The network was constructed using the interaction data from the TRRUST v2 database and the RNAInter v4.0 database. The circles represent hub genes. The triangles represent micro RNA. The square represents long non-coding RNA. The diamonds represent transcription factors. (**B**) Novel competing endogenous RNA subnetwork extracted from the whole regulatory network. Only nodes with high connectivity (degree > 5) were included.

## Data Availability

The original dataset analyzed in this study is publicly available. This data can be found here: https://www.ncbi.nlm.nih.gov/geo/query/acc.cgi?acc=gse73754, accessed on 11 June 2021. The raw data generated by secondary analysis of the dataset are available from the corresponding author, without undue reservation.

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
