# Peer review of "New Insights into the Regulatory Role of Ferroptosis in Ankylosing Spondylitis via Consensus Clustering of Ferroptosis-Related Genes and Weighted Gene Co-Expression Network Analysis"

_genes, 2022, doi:10.3390/genes13081373_

Round 1
Reviewer 1 Report
The manuscript presented by Tianhua Rong et al. entitled “New insights into the regulatory role of ferroptosis in ankylosing spondylitis via consensus clustering of ferroptosis-related genes and weighted gene co-expression network analysis” is not clear. Evene if the the topic is of great interest considering that Ankylosing spondylitis (AS) has a great impact on quality of life patients.
Major
In the methodological procedure, is not clear when patients are screened and how they received diagnosis of AS.
Besides that, please clarify why you decide to use ferroptosis related gene for this study
Author Response
In the methodological procedure, is not clear when patients are screened and how they received diagnosis of AS.
Response:
Thanks for your valuable comments. We checked the description of dataset GSE73754 (https://www.ncbi.nlm.nih.gov/geo/query/acc.cgi?acc=GSE73754) and referred to the corresponding citation. According to Gracey et al, all of the AS patients had radiographic sacroiliitis as defined by the modified New York criteria for AS, and were evaluated using the Bath Ankylosing Spondylitis Disease Activity Index. We have added relevant expressions in the “2.1 data collection” part.
Besides that, please clarify why you decide to use ferroptosis related gene for this study
Response:
Thanks for your insightful question. The reasons for using ferroptosis-related genes were mainly presented in the introduction part:
Firstly, Recent studies have increasingly reported on complex associations between ferroptosis and the immune system, and AS is also an autoimmune disease.
Secondly, Abnormal iron homeostasis occurs in rheumatic disease, and specifically, iron overload is observed in AS.
Thirdly, decreased levels of plasma thiol and antioxidant vitamins, increased lipid peroxidation, and protein oxidation in AS patients have been described. These findings correspond to the three pillars of ferroptosis, which are iron, thiols, and lipid peroxidation, collectively suggesting that ferroptosis participates in AS pathogenesis. We added the “three pillars” explanation to the text to enhance the logical connection between AS and ferroptosis.
Also, in the second paragraph of the Discussion part, echoing the introduction part, we mentioned that previous findings on ferroptosis-associated musculoskeletal disorders have led us to hypothesize that ferroptosis may also play a vital role in the development and progression of AS.
Last but not least, recent studies have also shown that ferroptosis plays a pathogenic role in inflammatory bowel disease (IBD). The theory of the gut–joint inflammatory axis postulates that IBD and AS are strongly interrelated, as exemplified by considerable epidemiological, clinical, genetic, and immunological overlaps. Therefore, the pivotal role of ferroptosis in IBD inspired us to speculate that ferroptosis may also play a role in AS. We didn’t write this into the text, given the scarcity of direct evidence.

Reviewer 2 Report
This interesting work gives information regarding ferroptosis-related genes in seronegative spondyloarthritis.
There are several issues to clarify to enrich the knowledge of communication.
The discussion about HLA-B27 should be more profound and include the information regarding the 52 AS patients included by adding basal clinical and demographic data of those patients if available; if not, recognize it as a limitation of the study.
The search terms included “seronegative spondyloarthritis,” an inclusive nosologic term that may include patients without classical AS displaying complementary immune mechanisms such as specific cytokine signatures. The lack of discussion IL-17 pathway is also a limitation of this study.
Author Response
The discussion about HLA-B27 should be more profound and include the information regarding the 52 AS patients included by adding basal clinical and demographic data of those patients if available; if not, recognize it as a limitation of the study.
Response:
Thanks for your valuable comments. In this study, the positive rate of HLA-B27 between the two groups showed no significant difference, while the proportion of CD8+ T cells differed between the two groups. Hence, in the discussion part, we paid less attention to HLA-B27 itself and focused on the presentation of arthritogenic peptides to CD8+ cytotoxic T cells. We added a sentence to the text describing the possible pathogenic mechanism of HLA-B27.
In this study, we used a publicly available dataset, which contained limited clinical information. We’ve tried our best but still can not get access to patients’ basal clinical and demographic data. We have pointed it out as a weakness in the last paragraph of the Discussion part.
The search terms included “seronegative spondyloarthritis,” an inclusive nosologic term that may include patients without classical AS displaying complementary immune mechanisms such as specific cytokine signatures. The lack of discussion IL-17 pathway is also a limitation of this study.
Response:
Thanks for your insightful question. When searching for eligible datasets, we used broader terms to avoid omitting any available data. Among the 14 retrieved results, 6 datasets that included analysis of peripheral blood samples obtained from AS patients were evaluated in detail. We then manually checked every dataset to confirm that the dataset contains patients with AS. We added the expression “All of the AS cases in this dataset had radiographic sacroiliitis as defined by the modified New York criteria for AS, and were evaluated using the Bath Ankylosing Spondylitis Disease Activity Index” to the “2.1 data collection” part.
The analysis results of this study had no direct connection with the IL-17 pathway. Thus, we merely mentioned IL-17 in passing when discussing the neutrophil extracellular traps induced ferroptosis. We can't agree more that the IL-23–IL-17 axis plays a pivotal role in AS. We added the lack of the IL-17 pathway to the limitation paragraph.
Reviewer 3 Report
In this paper, the authors reported their findings that implicate ferroptosis in the pathogenesis of ankylosing spondylitis (AS). By analysing AS samples from the Gene Expression Omnibus using consensus clustering of ferroptosis-related genes and weighted gene co-expression network analysis, the authors showed that AS patients can be divided into two subtypes with distinct ferroptosis-related genes (FRG) expression patterns. Furthermore, the authors also constructed a multi factor regulatory network for twelve hub genes. This paper is the first to speculate a potential role for ferroptosis in the pathogenesis of AS. Considering the aetiology of AS is unclear and yet to be elucidated, this paper does provide valuable insights into potential underlying mechanism for AS.
Overall, the paper is well written and the methods used were logically and clearly described. Just a few comments:
1) Is the sample size sufficient for this sort of study? There are significantly less control samples than AS samples.
2) While the authors did an excellent job describing the methods used in the analysis, there is relatively little information on the logic behind closing to study ferroptosis genes. It would be helpful if the authors also include more background, such as anaemia in AS patients, ferritin levels in patients etc
Author Response
- Is the sample size sufficient for this sort of study? There are significantly less control samples than AS samples.
Response:
Thanks for your valuable comments. According to Pawitan et al (False discovery rate, sensitivity and sample size for microarray studies. Bioinformatics. 2005 Jul 1;21(13):3017-24. PMID: 15840707), the required sample size in an experiment depends on (1) the number and (2) the distribution of the truly differentially expressed genes, and (3) on how much false discovery rate (FDR) we can tolerate. Usually, 20 samples per group is acceptable, and 30 samples or more is ideal. In this study, there were 24 and 28 samples in the two groups, respectively, which is sufficient for reaching robust analysis results.
In this study, we only used the AS samples in the GSE73754 dataset. The healthy control samples were not within the scope of this study.
- While the authors did an excellent job describing the methods used in the analysis, there is relatively little information on the logic behind closing to study ferroptosis genes. It would be helpful if the authors also include more background, such as anaemia in AS patients, ferritin levels in patients etc
Response:
Thanks for your insightful suggestion. The reasons for using ferroptosis-related genes were mainly presented in the introduction part:
Firstly, Recent studies have increasingly reported on complex associations between ferroptosis and the immune system, and AS is also an autoimmune disease.
Secondly, Abnormal iron homeostasis occurs in rheumatic disease, and specifically, iron overload is observed in AS.
Thirdly, decreased levels of plasma thiol and antioxidant vitamins, increased lipid peroxidation, and protein oxidation in AS patients have been described. These findings correspond to the three pillars of ferroptosis, which are iron, thiols, and lipid peroxidation, collectively suggesting that ferroptosis participates in AS pathogenesis. We added the “three pillars” explanation to the text to enhance the logical connection between AS and ferroptosis.
We also modified the corresponding sentence as “More specifically, iron overload, anemia, and altered ferritin level are observed in AS”.

Reviewer 4 Report
The manuscript by Tianhua Rong and colleagues reports the bioinformatical analysis of the microarray data set GSE73754 to explore a potential link between ferroptosis-related genes and Ankylosing spondylitis.
Using different bioinformatical tools, the analysis starts with 276 ferroptosis-related genes that were manually compiled. Consensus clustering is used to test how many subgroups are present in the GSE73754 dataset based on the 276 annotated genes. A set of 3663 differentially-regulated genes were identified between these two clusters although the differences are not significant. Amongst this group were 86 ferroptosis-related genes. Further analysis identified 12 hub genes amongst the 86 ferroptosis genes. Finally, data from a second database (TRRUST) was used to identify possible target genes under control of the 12 hub genes.
The author concluded that the two clusters of ferroptosis-related genes differ within the disease spectrum of Ankylosing spondylitis which correlates with changes in the immune cell background.
Main comment:
While the method section documents well the different steps, the reader is not informed about the rationale behind the analysis steps and the gene selection. The reader must be informed about the thinking that underpins the different steps. For example, it remains unclear on which basis the initial 276 ferroptosis-related genes were selected. This is of particular importance, since the remaining steps (i.e. cluster selection) are informed by the selected genes. Furthermore, it remains obscure how the 12 hub genes are linked with the 86 ferroptosis genes, how the 276 ferroptosis genes informed cluster selection, how the 12 hub genes were used to select the target genes in the TRRUST databank and how the 3663 DEGs relate to the 86 ferroptosis genes. This information may be hidden between the lines, but needs to be clearly spelled out in the Result section for the benefit of the reader.
I think that the authors cannot make a link between the selected ferroptosis genes and Ankylosing spondylitis yet as they analyse a gene set within the disorder simply based on the selected 276 ferroptosis genes of this ferroptosis gene cohort. The authors need to invalidate the following argument: if the same set of 276 ferroptosis-related genes were to be tested against a different immunological disorder (i.e. a different microarray data set), would this yield significant different hub genes? In other words, does the analysis simply reveal hub genes within the ferroptosis gene group?
Minor comments:
Please write for an audience which is not familiar with the technical terms of a bioinformatics analysis. For example, please explain how and why the 276 ferroptosis genes revealed 2 clusters. Please use the full expression when the abbreviations like DEGs, FRG, AS, WGCNA are used in the Result section. Please explain the significance of the turquoise module.
Figure 8: The expression difference of the 12 ferroptosis hub genes is not meaningful as the two clusters were chosen to be different in the first place based on the initial 276 ferroptosis genes. Hence, an expression difference is to be found.
The information from line 317-339 could move to the Introduction
Author Response
While the method section documents well the different steps, the reader is not informed about the rationale behind the analysis steps and the gene selection. The reader must be informed about the thinking that underpins the different steps. For example, it remains unclear on which basis the initial 276 ferroptosis-related genes were selected. This is of particular importance, since the remaining steps (i.e. cluster selection) are informed by the selected genes. Furthermore, it remains obscure how the 12 hub genes are linked with the 86 ferroptosis genes, how the 276 ferroptosis genes informed cluster selection, how the 12 hub genes were used to select the target genes in the TRRUST databank and how the 3663 DEGs relate to the 86 ferroptosis genes. This information may be hidden between the lines, but needs to be clearly spelled out in the Result section for the benefit of the reader.
Response:
Thanks for your valuable comments. To better inform the rationale of the analysis steps and the gene selection, we added a graphical abstract in the revised version of the manuscript, with a special focus on the process of obtaining the 276 FRGs. At the very beginning, we manually compiled ferroptosis-related genes (FRGs) from an online database FerrDb and a thorough literature review. The FerrDb contained 259 FRGs. By literature review, additional 41 FRGs were curated after removing duplication. Now we have a total of 300 FRGs. Then we annotated the probes in GEO microarray dataset GSE73754 with gene symbols and tried to locate the 300 FRGs in dataset GSE73754. Different microarray products contain different numbers and types of genes. After a thorough search, we identified 276 out of the 300 FRGs in this dataset. The other 24 FRGs were not contained by the present microarray.
The expression levels of the 276 FRGs were the foundation for consensus clustering. The algorithm “consensus clustering” is an unsupervised class discovery tool in genomic studies. It calculates how frequently two samples are grouped together in repeated clustering runs, and assesses cluster stability. This algorithm categorizes samples with similar expression patterns of the 276 FRGs into the same cluster. We added this expression to the text in the revised manuscript.
The 12 hub genes did not belong to the above-mentioned 276 FRGs and were not linked with the 86 differentially expressed FRGs. The 276 FRGs served as a classifier in the early stage of the analysis process. After the two clusters were obtained by consensus clustering algorithm based on the 276 FRGs, our scope of analysis shifted to the entire microarray, which contained more than 30 thousand probes. The 3663 DEGs were acquired by comparing the expression levels of all genes in the microarray. Among the 3663 DEGs, 86 genes belonged to the above-mentioned 276 FRGs. We modified the expression in the “3.3 Identification of DEGs between the two clusters” part according to your suggestion.
TRRUST (version 2, https://www.grnpedia.org/trrust/) is a manually curated database of transcriptional regulatory networks. The current version of TRRUST contains 8,444 and 6,552 TF-target regulatory relationships of 800 human TFs and 828 mouse TFs, respectively. They have been derived from 11,237 Pubmed articles, which describe small-scale experimental studies of transcriptional regulations. The TRRUST network edge information is freely available. We downloaded the entire TF-target information and searched for the transcription factor that regulates the 12 hub genes. One transcription factor, MTA1, was retrieved. The corresponding text is expressed in the “3.7 Multifactor regulatory network in AS” part.
Graphical abstract with a special focus on the process of obtaining the 276 ferroptosis-related genes. The arrows indicate the order of the analysis steps. AS, ankylosing spondylitis; FRGs, ferroptosis-related genes; DEG, differentially expressed gene; PCA, principal component analysis; WGCNA, weighted gene co-expression network analysis.
I think that the authors cannot make a link between the selected ferroptosis genes and Ankylosing spondylitis yet as they analyse a gene set within the disorder simply based on the selected 276 ferroptosis genes of this ferroptosis gene cohort. The authors need to invalidate the following argument: if the same set of 276 ferroptosis-related genes were to be tested against a different immunological disorder (i.e. a different microarray data set), would this yield significant different hub genes? In other words, does the analysis simply reveal hub genes within the ferroptosis gene group?
Response:
Thanks for your insightful comments and questions. As mentioned above, the 3663 DEGs were acquired by comparing the expression levels of all genes in the microarray. The 12 hub genes didn’t belong to the above-mentioned 276 ferroptosis-related genes. We added this clarification to the “3.6 Identification of hub genes” part.
Please write for an audience which is not familiar with the technical terms of a bioinformatics analysis. For example, please explain how and why the 276 ferroptosis genes revealed 2 clusters. Please use the full expression when the abbreviations like DEGs, FRG, AS, WGCNA are used in the Result section. Please explain the significance of the turquoise module.
Response:
Thanks for your valuable comments. As mentioned above, we have modified the expression in the “3.1 Consensus clustering based on FRGs” part to increase reader friendliness. After repeated discussions by the authors, we finally decided to provide more detailed background knowledge of consensus clustering as references 33-35. Your understanding will be more than appreciated.
According to the “Instructions for Authors” web page (https://www.mdpi.com/journal/genes/instructions): Acronyms/Abbreviations/Initialisms should be defined the first time they appear in each of three sections: the abstract; the main text; the first figure or table. When defined for the first time, the acronym/abbreviation/initialism should be added in parentheses after the written-out form. We were concerned that using the full expression when the abbreviations like DEGs, FRG, AS, WGCNA appear in the Result section would conflict with the journal’s requirements. Therefore, we only defined the abbreviations when they first appear in the main text, and consistently used the abbreviations thereinafter. Thanks for your generosity and understanding.
Figure 8: The expression difference of the 12 ferroptosis hub genes is not meaningful as the two clusters were chosen to be different in the first place based on the initial 276 ferroptosis genes. Hence, an expression difference is to be found.
Response:
Thanks for your insightful suggestions. As mentioned above, the 276 FRGs served as a classifier in the early stage of the analysis process. After the two clusters were obtained by consensus clustering algorithm based on the 276 FRGs, our scope of analysis shifted to the entire microarray, which contained more than 30 thousand probes. Thus, the 12 hub genes were among the 3663 DEGs, but did not belong to the 276 FRGs. Figure 8 served as a quantitative verification and visualization of the differential expression. The authors would be grateful if this figure could be retained.
The information from line 317-339 could move to the Introduction
Response:
Thanks for your instructive suggestion. In the Discussion part, the first paragraph aimed to reemphasize the significance of the present study. The second paragraph sought to explain the reason for choosing ferroptosis as the research topic from an orthopedic perspective. The sentences in line 317-339 were intentionally put in this place of the article, echoing the Introduction part. In the Introduction part, the penultimate paragraph explained the reason for choosing ferroptosis as the research topic from a rheumatological perspective. The purpose and significance of this study were pointed out for the first time in the last paragraph. The authors would like to keep this structure unchanged if possible. Your understanding will be more than appreciated.

Round 2
Reviewer 1 Report
The authors responded to the reviewers' comments and made the suggested changes.
Author Response
Thanks for your valuable comments. Your recognition is a great encouragement to us.
Reviewer 4 Report
Dear authors,
thank you very much for your thorough responses to my comments.
Here are last recomendations:
Please include the text from your reply "At the very beginning, we manually compiled ferroptosis-related genes (FRGs) from an online database FerrDb and a thorough literature review. The FerrDb contained 259 FRGs. By literature review, additional 41 FRGs were curated after removing duplication. Now we have a total of 300 FRGs. Then we annotated the probes in GEO microarray dataset GSE73754 with gene symbols and tried to locate the 300 FRGs in dataset GSE73754. Different microarray products contain different numbers and types of genes. After a thorough search, we identified 276 out of the 300 FRGs in this dataset. The other 24 FRGs were not contained by the present microarray." at the start of the Result section.
Please include "The current version of TRRUST contains 8,444 and 6,552 TF-target regulatory relationships of 800 human TFs and 828 mouse TFs, respectively. They have been derived from 11,237 Pubmed articles, which describe small-scale experimental studies of transcriptional regulations." in Line 296 where the TRRUST data base is introduced.
Pease include "After the two clusters were obtained by consensus clustering algorithm based on the 276 FRGs, our scope of analysis shifted to the entire microarray, which contained more than 30 thousand probes. Thus, the 12 hub genes were among the 3663 DEGs, but did not belong to the 276 FRGs." at the beginning of 3.3. Identification of DEGs between the two clusters
Author Response
Please include the text from your reply "At the very beginning, we manually compiled ferroptosis-related genes (FRGs) from an online database FerrDb and a thorough literature review. The FerrDb contained 259 FRGs. By literature review, additional 41 FRGs were curated after removing duplication. Now we have a total of 300 FRGs. Then we annotated the probes in GEO microarray dataset GSE73754 with gene symbols and tried to locate the 300 FRGs in dataset GSE73754. Different microarray products contain different numbers and types of genes. After a thorough search, we identified 276 out of the 300 FRGs in this dataset. The other 24 FRGs were not contained by the present microarray." at the start of the Result section.
Response:
Thanks for your valuable suggestion. Regarding our previous reply, “At the very beginning, we manually compiled ferroptosis-related genes (FRGs) from an online database FerrDb and a thorough literature review. The FerrDb contained 259 FRGs. By literature review, additional 41 FRGs were curated after removing duplication. Now we have a total of 300 FRGs”, the corresponding description has been written in the “2.2 Ferroptosis-related genes” part.
As for the latter half, “Then we annotated the probes in GEO microarray dataset GSE73754 with gene symbols and tried to locate the 300 FRGs in dataset GSE73754. Different microarray products contain different numbers and types of genes. After a thorough search, we identified 276 out of the 300 FRGs in this dataset. The other 24 FRGs were not contained by the present microarray”, we refined the language and added it at the beginning of the Results part.
Please include "The current version of TRRUST contains 8,444 and 6,552 TF-target regulatory relationships of 800 human TFs and 828 mouse TFs, respectively. They have been derived from 11,237 Pubmed articles, which describe small-scale experimental studies of transcriptional regulations." in Line 296 where the TRRUST data base is introduced.
Response:
Thanks for your helpful suggestion. Since these sentences are not the result of the present study, the authors decided to add it in the Method part, more specifically, the “2.9 Construction of regulatory network based on hub genes” part.
Pease include "After the two clusters were obtained by consensus clustering algorithm based on the 276 FRGs, our scope of analysis shifted to the entire microarray, which contained more than 30 thousand probes. Thus, the 12 hub genes were among the 3663 DEGs, but did not belong to the 276 FRGs." at the beginning of 3.3. Identification of DEGs between the two clusters.
Response:
Thanks for your insightful suggestion. Regarding our previous reply, “After the two clusters were obtained by consensus clustering algorithm based on the 276 FRGs, our scope of analysis shifted to the entire microarray, which contained more than 30 thousand probes”, we add it to the “3.3 Identification of DEGs between the two clusters” part as suggested.
As for the sentence, “Thus, the 12 hub genes were among the 3663 DEGs, but did not belong to the 276 FRGs”, the corresponding expression has been added to the “3.6 Identification of hub genes” part.